# Antimicrobial Resistance and Molecular Characterization of Methicillin-Resistant *Staphylococcus aureus* Isolated from Slaughtered Pigs and Pork in the Central Region of Thailand

**DOI:** 10.3390/antibiotics10020206

**Published:** 2021-02-19

**Authors:** Wimonrat Tanomsridachchai, Kanjana Changkaew, Ruchirada Changkwanyeun, Watsawan Prapasawat, Apiradee Intarapuk, Yukari Fukushima, Nattapong Yamasamit, Thoko Flav Kapalamula, Chie Nakajima, Orasa Suthienkul, Yasuhiko Suzuki

**Affiliations:** 1Division of Bioresources, Research Center for Zoonosis Control, Hokkaido University, Sapporo 001-0020, Japan; wimonrat@czc.hokudai.ac.jp (W.T.); yukaring@czc.hokudai.ac.jp (Y.F.); thokokapalamula@czc.hokudai.ac.jp (T.F.K.); cnakajim@czc.hokudai.ac.jp (C.N.); 2Faculty of Public Health, Thammasat University (Rangsit Campus), Pathumthani 12121, Thailand; k_changkaew@hotmail.com (K.C.); nuttapong.y@fph.tu.ac.th (N.Y.); orasa.s@fph.tu.ac.th (O.S.); 3Faculty of Public Health, Thammasat University (Lampang Campus), Lampang 52190, Thailand; ruchirada.c@fph.tu.ac.th; 4Department of Clinic, Faculty of Veterinary Medicine, Mahanakorn University of Technology, Bangkok 10530, Thailand; pigfats@hotmail.com (W.P.); aintarapuk@hotmail.com (A.I.); 5International Collaboration Unit, Research Center for Zoonosis Control, Hokkaido University, Sapporo 001-0021, Japan; 6Faculty of Public Health, Mahidol University, Bangkok 10400, Thailand

**Keywords:** methicillin-resistant *Staphylococcus aureus*, antimicrobial resistance, genotype, slaughtered pigs, pork

## Abstract

Methicillin-resistant *Staphylococcus aureus* (MRSA) have been a major public health concern in humans. Among MRSA, livestock-associated (LA)-MRSA strains have always been associated with exposure to livestock or their products and have emerged in different countries globally. Although studies have identified LA-MRSA from healthy pigs and pork in Thailand, prevalence in slaughtered pigs is still unknown. In addition, there are few reports on the epidemiology and molecular characteristics of LA-MRSA in Thailand. Hence, this is the first report investigating the epidemiology and molecular characteristics of MRSA in individual slaughtered pigs and pork in Thailand. A total of 204 nasal swab and 116 retailed pork samples were collected from three slaughterhouses and four fresh markets, respectively. Individual samples were used for screening for MRSA and obtained isolates were examined for drug- resistance profiling for 12 antimicrobial agents of 10 drug classes. In addition, SCC*mec* typing and multi-locus sequence typing were conducted to obtain genotype profiles. MRSA were isolated from 11 and 52 nasal swab and pork samples, respectively. The prevalence was significantly higher in the pork than in the nasal swab samples (*p*-value < 0.05). A high prevalence of ST9-SCC*mec*IX and ST398-SCC*mec*V with high-level antimicrobial resistance from markets and slaughterhouses indicated the spreading of MRSA with these genotypes in the Thai swine processing chains and suggested the need for further investigation to determine a control.

## 1. Introduction

Methicillin-resistant *Staphylococcus aureus* (MRSA) has been a major public health concern as it causes nosocomial infections leading to high mortality and morbidity in humans [1]. MRSA strains with resistance to a wide range of antibiotics have been found in various sources globally [2,3,4] and are classified into three broad groups, namely, healthcare-associated MRSA (HA-MRSA), community-associated MRSA (CA-MRSA), and livestock-associated MRSA (LA-MRSA) based on the source of infection [5]. The majority of MRSA infections originate from hospitals and other healthcare facilities and spread into the community [2]. CA-MRSA has been increasingly reported in communities among people without prior history of risk factors to MRSA infections [6]. LA-MRSA strains have always been associated with exposure to livestock or their products and have emerged in different countries in Europe, America, and Asia [7,8]. The most prominent LA-MRSA strain in Europe and America belongs to sequence type 398 (ST398) whereas that in most Asia countries belongs to ST9 [9,10,11,12,13]. MRSA types have divergent genetic backgrounds, hence different MRSA strains carry different types of staphylococcal cassette chromosome mec (SCC*mec*) [7]. LA-MRSA belonging to ST398 has been reported to colonize livestock and people with close contact to them such as farmers and veterinarians [14,15,16].

However, infections by LA-MRSA were also found in people without livestock exposure [17,18]. ST398 and several others (ST9, ST97, ST5) have been isolated from pork, chicken, beef, and milk in many countries [19]. These finding demonstrate that handling and/or consumption of food-producing animals contaminated by MRSA is a potential zoonotic transmission source for humans [20,21]. When MRSA-carrying animals are slaughtered, MRSA may spread to carcasses, to the environment, and to abattoir workers. Moreover, if animal products are contaminated, MRSA can enter the human food chain [6]. Therefore, LA-MRSA has become an important public health issue that warrants intensive monitoring.

Thailand has a positive trend for the production and export of pork and live pigs especially to ASEAN countries and domestic pork consumption increased 2–3% from 2011 to 2016 [22]. As the central region of Thailand is the main pig production area [23], the risk of zoonotic transmission of LA-MRSA through pigs and/or pig products is high [24]. Although some studies have identified LA-MRSA from healthy pigs [13,25], and pork [26] in Thailand, the prevalence in slaughtered pigs is still unknown. Moreover, there is only one report on the description of the epidemiology and molecular characteristics of LA-MRSA from slaughtered pigs and pork in Thailand [7]. Therefore, the purpose of this study was to investigate the prevalence, molecular characteristics, and antimicrobial resistance pattern of MRSA isolated from slaughtered pigs and retail pork in the central region of Thailand.

## 2. Results

### 2.1. Prevalence of MRSA 

Among 204 nasal swab samples of pigs from three slaughterhouses and 116 pork samples from four markets, 63 (19.7%) were positive for MRSA based on the presence of the *mecA* (Table 1). The prevalence was significantly higher in pork samples (44.8%; 52/116) than in nasal swab samples (5.4%; 11/204) (*p*-value < 0.05) (Table 1 and Appendix A). No MRSA was found in nasal swab samples from slaughterhouse C in both year (2017 and 2018) or in pork samples from market D in the first year (2017). Among nasal swab samples, the highest prevalence of MRSA was found at slaughterhouse A (11.8%; 8/68). For pork samples, the highest prevalence of MRSA was found at market F (58.3%; 7/12) followed by market G (50.0%; 8/16), market E (48.0%; 36/75), and market D (7.7%; 1/13). There were no significant differences between the sampling years (Table 1). In total, 67 MRSA isolates, 11 from nasal swab and 56 from pork samples, were used for further analyses (Table 1 and Table 2).

### 2.2. Antimicrobial Susceptibility

Drug susceptibility tests utilizing 12 antimicrobial agents of 10 drug classes revealed that all examined isolates were resistant to ampicillin and cefoxitin, and various degrees of resistance were observed in other 10 antimicrobial agents with all isolates susceptible to vancomycin as shown in Appendix A. There was no statistically significant difference between nasal swab and pork samples in the prevalence of each antimicrobial resistance (Fisher’s test; *p*-value > 0.05). All MRSA isolates were multi-drug resistant (MDR) and classified into 18 different patterns of resistance (Appendix A). Six and 16 different patterns of drug resistance were observed in isolates from nasal swabs and pork samples, respectively. Nevertheless, it was found that all of isolates showed resistance to at least two of the non-β-lactams antimicrobial classes. All isolates from nasal swab samples were MDR, resulting in resistance to at least three non-β-lactams antimicrobial classes, whereas only 39 (69.6%) MRSA isolates from pork samples were MDR. The antimicrobial resistance profile of ampicillin-oxacillin-cefoxitin-clindamycin-tetracycline (AMP-OXA-FOX-CLI-TET), was the highest in frequency (23.9%; 16/67) and found only in pork samples from Market E (in 2017) and market F (both 2017 and 2018), followed by AMP-OXA-FOX-chloramphenicol(CHL)-CLI-enrofloxacin(ERY)-ciprofloxacin(CIP)-erythromycin(ENR)-gentamicin(GEN)-TET (16.4%; 11/67) and AMP-OXA-FOX-CHL-CLI-CIP-ENR-GEN-TET (16.4%; 11/67) found in both nasal and pork samples. The other antimicrobial resistance patterns, which were mainly found in pork samples for both years, were diverse and low in number.

### 2.3. Molecular Characteristics (by Multilocus Sequence Typing (MLST) and SCCmec Typing) of MRSA Isolates

MRSA isolates were differentiated into four SCC*mec* types and four STs. SCC*mec* type IX was the most prevalent (68.7%; 46/67), followed by SCC*mec* type V (26.9%; 18/67) and SCC*mec* type IV (1.5%; 1/67), while two isolates (3.0%), consisting of class C2 *mec* complex but negative amplification of ccr complex were nontypeable (NT). The most frequently found ST was ST9 (70.1%; 47/67) followed by ST398 (26.9%; 18/67), ST779 (1.5%; 1/67), and ST5639 (1.5%; 1/67) (Table 3). ST5639 was a new single-locus variant of ST9 with a substitution mutation (G52T) of *glpF*, resulting in allelic profile 3-3-111-1-1-1-10, which belonged to clonal complex (CC)9. The five different genotype profiles were identified where ST9-SCC*mec* IX was predominant in both nasal swabs and pork samples. ST398-SCC*mec* V was identified at market F (in both years), market E (only in the first year), and at slaughterhouse A (only in the first year). MRSA at market E in the first year (2017) showed the most diverse molecular characteristic profiles (Table 2). Four samples were found to carry two strains with different genotype profiles in each. The characteristic genotype profile of ST9-SCC*mec* IX and ST398-SCC*mec* V were found in a pork sample from market E and two pork samples from market F. Moreover, ST398-SCC*mec* V and ST9-SCC*mec* NT were found in a pork sample from market E.

### 2.4. Association between Antimicrobial Resistance and Molecular Typing

Antimicrobial resistance rates obtained for five different genotype profiles are shown in Table 3 and Figure 1. ST9-SCC*mec* IX isolates showed significantly higher rates of resistance (*p*-value < 0.05) than isolates with other genotype profiles, exhibiting high prevalence of resistance to chloramphenicol, erythromycin, ciprofloxacin, enrofloxacin, gentamicin, and sulfamethoxazole/trimethoprim. Among ST398-SCC*mec* V isolates (n = 18), the antimicrobial resistance pattern AMP-OXA-FOX-CLI-TE was found with the highest frequency in pork samples from markets (88.9%; 16/18) (Figure 1 and Appendix A). All MRSA ST398-SCC*mec* V from market F in both years exhibited the same antimicrobial resistance profile, whereas one MRSA isolate from market E in the first year was different in antimicrobial resistance pattern.

## 3. Discussion

This study investigated the distribution of MRSA in individual slaughtered pigs and pork in markets at central region of Thailand. This is the first report investigating the epidemiology and molecular characteristics of MRSA in individual slaughtered pigs and pork in Thailand.

The prevalence of MRSA in nasal swab samples observed in this study (11/204; 5.4%) was lower than that in European countries such as Latvia (17/100; 17%) [27] and other Asian countries such as China (38/590; 6.4%) [28]. We estimated the prevalence of MRSA isolated from pork as 44.8% (52/116), which is slightly lower than that in the earlier study in the central region of Thailand (50%;5/10) [26]. In contrast, our study results were higher than 1.8–15.8% among pork in European countries [29], 3.6–9.6% in North American countries [30,31], and 7.1–21.5% in some Asian countries [32,33]. The prevalence may vary depending on several factors, such as geographical area, sampling methods, sample size, collection period, and laboratory methodologies.

In our study, the frequently-observed STs were ST9 and ST398 which are known to be associated with animals. These are major endemic MRSA clones circulating in pigs in the central region of Thailand [24,26]. ST9 represents the most common sequence type in Asian countries [7] while ST398 is the dominant clone disseminating worldwide, especially in Europe and North America [3], and has been rarely identified in some Asian countries [34]. However, these strains are an infection-associated strain among pigs and humans in other Asian countries [7,34]. Although ST398 strains have been found from veterinarian [35] and swine farms (pigs and swine workers) [24] in Thailand, our report is the first to detect this strain from pork samples. ST9 and ST398 might be endemic in animal food production in the central region of Thailand.

ST9 has been rarely associated with human diseases [3]; however, a report from Thailand identified ST9 in 2.5% (7/276) of pig farm workers’ isolates [36]. In our study, ST5639, a novel single-locus variant of ST9, with a single base substitution in *glpF* was detected in pork from the market. This finding supports the notion that pigs or food animals are reservoirs for the emergence of new MRSA lineages or the evolution of existing clones [37]. Of note, one ST779 isolate from pork in the market (Table 2) was closely related to CA-MRSA or HA-MRSA observed among the population in Australia, UK, Ireland, and France [38,39]. We detected MRSA ST779 clone carrying SCC*mec* type IV (Table 2) distinct from a previous report by Kinnevey et al. [40,41] and Roberts et al. [42]. The former and the latter found ST779 carrying pseudo-element ΨSCC*mec*-SCC-SCC_CRISPR_ and SCC*mec* type V, respectively. The emergence of human-related STs indicates that slaughter pigs and pork could become important reservoirs for MRSA and increase the potential risk of human infections. Thus, the MRSA lineage described in this study should be considered as a possible public health threat. These data suggest the need to investigate production practices in farms supplying pork products to markets.

A high prevalence of SCC*mec* IX and V among MRSA isolates from markets and slaughterhouses (Table 2 and Figure 1) indicates that this MRSA genotype is rapidly spreading among swine processing chain. ST9 isolates carry different types of SCC*mec* depending on the country [12,13,28,43]. Moreover, a large variety of SCC*mec* types have been found in CC9 strains; much more so than CC398 stains [8]. Therefore, the structures of the non-typeable SCC*mec* found in ST9 in this study need to be characterized by whole-genome sequencing in our future study.

We discovered a high diversity of MRSAs genotypes in markets. The major genotype profiles of MRSA isolates were different in each year and each source (slaughterhouses and markets). This analysis suggests that it may be linked to multiple sources of pork in each market and to a temporal shift in the epidemiology of genotype (STs and SCC*mec* type) in Thailand. Hence, further study is needed to monitor the evolution of these pathogens among livestock especially in pig farms and food production stages. Moreover, investigations of LA-MRSA compared to HA-MRSA and CA-MRSA in the same area should be conducted to elucidate the source of cross-contamination of MRSA among the human population, since certain clones may spread in this population.

Notably, oxacillin-susceptible *mecA*-positive *S. aureus* (OS-MRSA) found in one pork-sample isolate belonged to ST9-SCC*mec* IX (Table 3 and Figure 1). All of LA-MRSA ST398 (Table 3 and Figure 1) displayed resistance to tetracycline similar to the previous reports [44]. This demonstrates that the LA-MRSA ST398 strain originated as methicillin-susceptible *S. aureus* in humans, then acquired methicillin and tetracycline resistance by antimicrobial selective pressure within the pig farms [45]. Thus, human exposure to LA-MRSA ST 398 might lead to the readaptation of this clone by reacquisition of human pathogenicity genes [45,46]. The MRSA ST9 strain showed more diverse antimicrobial resistance profiles than ST398 clones. Similar profiles to ST9 have been reported in central Thailand [24]. Previous reports have shown that LA-MRSA isolates were resistant to at least one agent of the fluoroquinolone class in Thailand [13,24,26,36,47,48,49], whereas only LA-MRSA ST9 in our study was associated with fluoroquinolone resistance. It is possible that several fluoroquinolones are available for treatment of animals in farms, and thus, their use may increase resistance among LA-MRSA. These results indicated that appropriate use of antimicrobials in farms is necessary to avoid emergence of high antimicrobial resistance rates of MRSA which can be sources of transmission to humans via food and other routes.

## 4. Materials and Methods

### 4.1. Study Design and Sample Collection

The cross-sectional study was performed in two settings of the food chain—slaughterhouses and markets in the central region of Thailand in 2017 and 2018—to determine the prevalence of MRSA. 

A total of 204 nasal swab samples were collected from three slaughterhouses (A, B, and C) during 2017–2018 (Figure 2). In each year, 34 nasal swab samples were collected from each of the three slaughterhouses. All slaughterhouses were under the control of Department of Livestock Development, Ministry of Agriculture and Cooperatives, but under different ownerships. Slaughterhouse A belonged to the town-municipal while slaughterhouses B and C belonged to private companies. Approximately 100–200 pigs were slaughtered per day. Slaughtering of animals was done according to common slaughtering practice, nasal swab samples were collected immediately after the scalding and dehairing and prior to washing the head with water. A cotton swab was inserted 2–7 cm (according to swine size) into both nostrils and gently rotated against the mucosal epithelium. Then, the cotton swab was inserted in the tube containing medium (Seed Swab γ No. 2 “Eiken”; Eiken Chemical, Tokyo, Japan) and the cap was tightly closed. All the swab samples were immediately stored in an ice box.

Approximately 50 g raw pork samples were purposely purchased from each butcher shop and collected in individual plastic bags. A total of 116 retailed pork samples were collected from 64 butcher shops in four fresh markets (D, E, F, and G in Figure 2) in the 2-year study. In 2017, a total of 57 pork samples were collected from 32 butcher shops, including market D (*n* = 6), market E (*n* = 37), market F (*n* = 6), and market G (*n* = 8). In 2018, a total of 59 pork samples were collected from 32 butcher shops, including market D (*n* = 7), market E (*n* = 38), market F (*n* = 6), and market G (*n* = 8). The unequal number of butcher shops for sample collection in each market was dependent on the capacity of the market.

Slaughterhouses and fresh markets were selected for convenience, based on the willingness of the producers to participate. All samples were kept individually in sterile bags, stored in an icebox, and transported to the laboratory within 6 h for further processing.

This study used meat and carcass from pigs in markets and slaughterhouses that had been legally registered. The Institutional Animal Care and Use Committee, Thammasat University (IACUC-TU) has confirmed that no ethical approval is required.

### 4.2. Isolation and Identification of MRSA

All samples were inoculated into trypticase soy broth (TSB; Oxoid, Basingstoke, United Kingdom) containing ceftizoxime (5 ug/mL), aztreonam (75 mg/mL), and 6.5% NaCl, and incubated at 37 °C for 24 h. Subsequently, enrichment cultures from individual samples were streaked on oxacillin-resistance screening agar (ORSA) supplemented with 2 µg/mL oxacillin (Oxoid) and incubated at 37 °C for 24–48 h. Up to three suspected staphylococcal colonies (mannitol-positive) were selected per sample from ORSA and sub-cultured on trypticase soy agar (TSA) (Oxoid). Colonies on TSA were primarily identified by Gram stain, catalase test, coagulase test, DNase test, and growth on mannitol salt egg-yolk agar (Appendix A). Presumptive MRSA isolates were further confirmed to species level by sequencing of 16S rRNA gene using primers Bact-rrs-F (5′-AGAGTTTGATCCTGGCTCAG-3′) and Bact-rrs-R (5′- TACGGCTACCTTGTTACGAC-3′) [50]. The PCR reaction mixture (total 20 µL) consisted of 1× Ex *Taq* buffer, 1 mM MgCl_2_, 0.25 mM of each dNTP, 0.25 µM of each primer, 0.5 U *Taq* polymerase (Takara Bio Inc., Kyoto, Japan), and 1 µL of DNA template. Thermal cycling was performed in a Thermal Cycler (Applied Biosystems Veriti™ Thermal Cycler, Foster City, CA, USA). Amplification conditions entailed the following: initial denaturation at 96 °C for 1 min, 35 cycles of denaturation at 96 °C for 10 s, annealing at 55 °C for 10 s, DNA extension at 72 °C for 30 s, and final extension at 72 °C for 5 min. This protocol was adapted from Neilan et al. (1997) [50]. After sequencing of the 16S rRNA gene, contiguous sequences were analyzed by the BLAST search engine (http://www.ncbi.nih.gov accessed on 19 February 2021) and compared with those registered in the GenBank database.

Detection of the *mecA* gene was done by PCR using specific primers *mecA*-F (5′-AAAATCGATGGTAAAGGTTGGC-3′) and *mecA*-R (5′- AGTTCTGCAGTACCGGATTTGC-3′) for methicillin-resistance confirmation [51]. The PCR mixture was prepared in a total volume of 20 µl per reaction. The mixture contained 1x Green Go*Taq* reaction buffer, 1 mM MgCl_2_, 0.25 mM each of dNTP, 0.25 µM of each primer, 0.5 U GoTaq DNA polymerase (Promega, Madison, WI, USA), and 1 µl of DNA template. The final volume was adjusted to 20 µl with sterile deionized water. The PCR conditions were the same as explained in the previous study [51]. Isolates with *mecA* were kept frozen at −80 °C until further examination.

### 4.3. Antimicrobial Susceptibility Testing (AST)

Isolates identified as MRSA were examined for susceptibility to antimicrobial agents using the disk diffusion method on Mueller–Hinton agar (MHA; Oxoid) following the Clinical and Laboratory Standards Institute (CLSI) guidelines CLSI VET01 S5, 2018 for enrofloxacin [52]; and CLSI M100 S30, 2020 for all other antibiotics [53]. A total of 12 antimicrobial disks were used comprised of ampicillin (AMP, 10 µg), oxacillin (OXA, 1 µg), cefoxitin (FOX, 30 µg), chloramphenicol (CHL, 30 µg), clindamycin (CLI, 2 µg), erythromycin (ERY, 15 µg), ciprofloxacin (CIP, 5 µg), enrofloxacin (ENR, 5 µg), gentamicin (GEN, 10 µg), tetracycline (TET, 30 µg), sulfamethoxazole/trimethoprim (SXT, 25 µg), and vancomycin (VAN, 30 µg).

### 4.4. Molecular Typing of MRSA

SCC*mec* typing of MRSA was performed by PCR amplification of the *mec* (classes A–C) and *ccr* (types 1, 2, 3, and 5) regions as previously described [54]. The combinations of *ccr* types and classes of *mec* gene complexes were used to determine the SCC*mec* types of each isolate.

Multilocus sequence typing (MLST) was performed following the protocol described elsewhere [55]. The seven housekeeping genes (*arcC*, *aroE*, *glpF*, *gmk*, *pta*, *tpi*, and *yqi*) were amplified by PCR. After agarose gel electrophoretic separation, PCR products were purified using ExoSAP-IT™ PCR Product Cleanup Reagent (Thermo Fisher Scientific, Waltham, MA, USA). The concentration and quality of the purified PCR products were measured by Qubit 3 using Qubit dsDNA HS (High Sensitivity) Assay Kit (Thermo Fisher Scientific). The purified products were sequenced by ABI 3500xL Genetic Analyzer (Thermo Fisher Scientific) using a BigDye ver. 3.1 Terminator Cycle Sequencing Kit (Thermo Fisher Scientific). The sequencing data were analyzed using BioEdit version 7.0.9.1 [56]. The allele numbers and sequence type (ST) of each *S. aureus* isolate were obtained using MLST Databases *S. aureus* database (http://saureus.mlst.net accessed on 19 February 2021). Phylogenetic trees were constructed by Molecular Evolutionary Genetics Analysis (MEGA) software version 6.0 (www.megasoftware.net accessed on 19 February 2021). Isolates showing identical antimicrobial resistance phenotype and genotype obtained from same sample were considered as clonal.

### 4.5. Data Analysis

The SPSS software version 19.0 was used for statistical analysis. The chi-square tests or Fisher’s exact tests were carried out to examine the differences in the prevalence of MRSA and antimicrobial resistance profiles among the MRSA isolates. The *p*-value less than 0.05 was considered statistically significant.

## 5. Conclusions

This is the first report investigating the distribution of MRSA in individual slaughter pigs and pork in Thailand. A high prevalence of SCC*mec* IX and V with high-level antimicrobial resistance among MRSA isolates from markets and slaughterhouses indicated that MRSA with this genotype was rapidly spreading in Thai swine-processing chains. For planning countermeasures, further research is required to understand the nationwide epidemiology of LA-MRSA among livestock, especially in pig farms and food production. In accordance with the information obtained from this study, reduced usage of antimicrobials in farms, prevention of MRSA contamination in animals along the entire pig production chain, and improved hygiene in food practices can be recommended to control the spread of MRSA and reduce the risk of MRSA to a minimum.

## Figures and Tables

**Figure 1 antibiotics-10-00206-f001:**
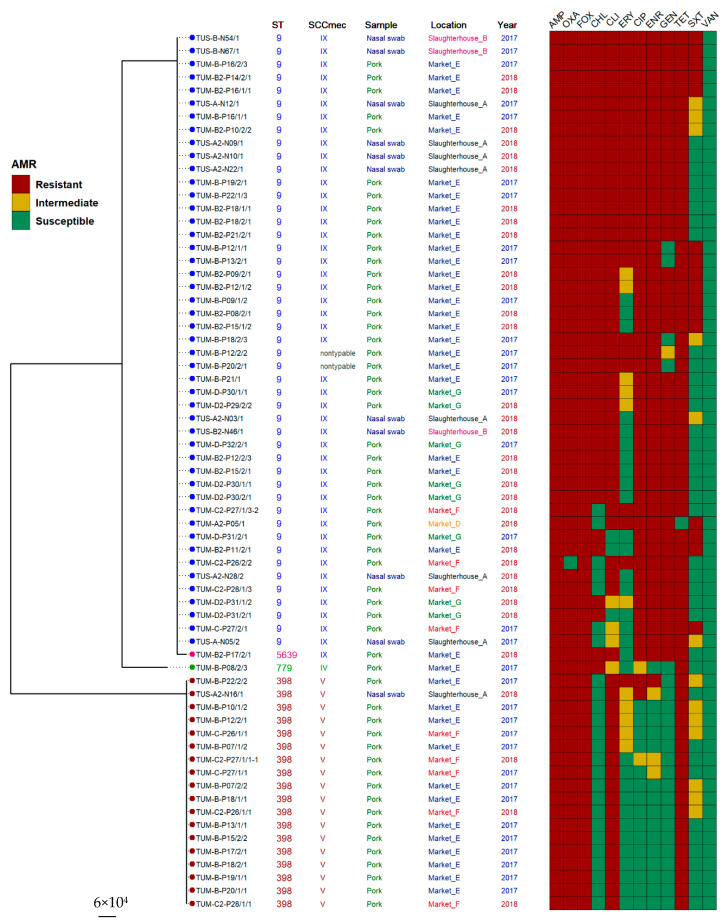
Phylogenetic tree showing the relationship between 67 MRSA strains isolated from nasal swab and pork samples based on the concatenated sequences of seven housekeeping enzyme genes’ loci (3186 bp). Boxes showing resistant, dark red; intermediate, ocher; and susceptible, green.

**Figure 2 antibiotics-10-00206-f002:**
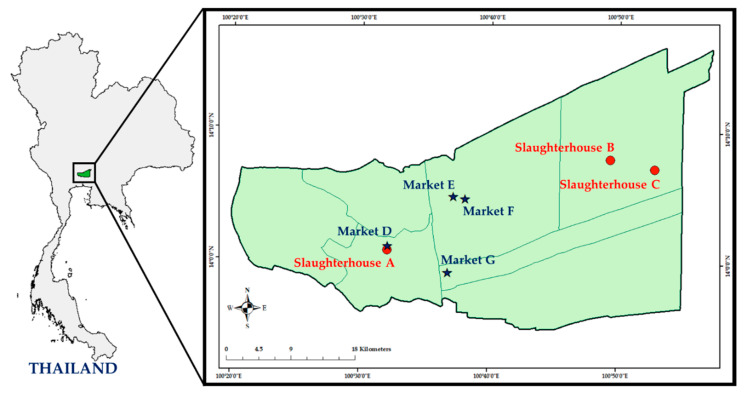
Geographic distribution of the three selected slaughterhouses and the four fresh markets in the central region of Thailand.

**Table 1 antibiotics-10-00206-t001:** Methicillin-resistant *Staphylococcus aureus* (MRSA) isolated in slaughterhouses and markets located in the central region of Thailand in 2017 and 2018.

Sample/Place	No. of MRSA-Positive Samples/Total No. (%)		
	2017	2018	Total
**Nasal swab**			
Slaughterhouse A	2/34 (5.9)	6/34 (17.6)	8/68 (11.8)
Slaughterhouse B	2/34 (5.9)	1/34 (2.9)	3/68 (4.4)
Slaughterhouse C	0/34	0/34	0/68
**Total (*n*)**	**4/102 (3.9)**	**7/102 (6.9)**	**11/204 (5.4)**
**Pork**			
Market D	0/6	1/7 (14.3)	1/13 (7.7)
Market E	22/37 (59.5) ^a^	14/38 (36.8)	36/75 (48.0)
Market F	3/6 (50.0)	4/6 (66.7) ^a^	7/12 (58.3)
Market G	3/8 (37.5)	5/8 (62.5)	8/16 (50.0)
**Total (*n*)**	**28/57 (49.1)**	**24/59 (40.7)**	**52/116** **(44.8)**

^a^ Two MRSA isolates were derived from one sample (there were two samples).

**Table 2 antibiotics-10-00206-t002:** Characteristics of staphylococcal cassette chromosome mec (SCC*mec)* type and ST type of MRSA isolated in slaughterhouses and markets located in the central region of Thailand in 2017 and 2018.

Typing Profiles		Slaughterhouse						Market								Total
		2017			2018			2017				2018				
SCC*mec* Typing	ST	A	B	C	A	B	C	D	E	F	G	D	E	F	G	
**IX**	**9**	**2**	**2**	0	**5**	**1**	0	0	**9**	**1**	**3**	**1**	**13**	**3**	**5**	**45**
**V**	**398**	0	0	0	**1**	0	0	0	**12**	**2**	0	0	0	**3**	0	**28**
**NT**	**9**	0	0	0	0	0	0	0	**2**	0	0	0	0	0	0	**2**
**IV**	**779**	0	0	0	0	0	0	0	**1**	0	0	0	0	0	0	**1**
**IX**	**5639**	0	0	0	0	0	0	0	0	0	0	0	**1**	0	0	**1**

NT, nontypeable; ST, sequence type.

**Table 3 antibiotics-10-00206-t003:** Association between antimicrobial resistance and genotype profile.

Genotype Profiles		Antimicrobial Agents (No. of Isolates)											
		AMP	OXA	FOX	CHL	CLI	ERY	CIP	ENR	GEN	TET	SXT	VAN
**ST9-SCC*mec* IX**	**(*n* = 45)**	45	44	45	**38 ***	**39**	**22 ***	**45 ***	**45 ***	**42 ***	**44**	**16 ***	0
**ST398-SCC*mec* V**	**(*n* = 18)**	18	18	18	0	18	1	2	1	0	18	0	0
**ST9-SCC*mec* NT**	**(*n* = 2)**	2	2	2	2	2	2	2	2	0	2	0	0
**ST779-SCC*mec* IV**	**(*n* = 1)**	1	1	1	1	0	0	0	0	0	1	0	0
**ST5639-SCC*mec* IX**	**(*n* = 1)**	1	1	1	1	1	0	1	1	1	1	0	0

Abbreviations: AMP, ampicillin; OXA, oxacillin; FOX, cefoxitin; CHL, chloramphenicol; CLI, clindamycin; ERY, erythromycin; CIP, ciprofloxacin; ENR, enrofloxacin; GEN, gentamicin; TET, tetracycline; SXT, sulfamethoxazole/trimethoprim; VAN, vancomycin; resistant: only resistant isolates, non-resistant: including susceptible and intermediate isolates; NT, nontypeable; significant differences between ST9-SCC*mec* IX and all other genotype profiles; * *p*-value less than 0.05 was considered statistically significant.

## Data Availability

Not applicable.

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
