# Peer review of "Antimicrobial Resistance and Molecular Characterization of Methicillin-Resistant Staphylococcus aureus Isolated from Slaughtered Pigs and Pork in the Central Region of Thailand"

_antibiotics, 2021, doi:10.3390/antibiotics10020206_

Round 1

Reviewer 1 Report

The manuscript by "Tanomsridachchai et al" investigated the molecular characteristics, epidemiology and antimicrobial resistance patterns of MRSA in individual slaughter pigs and pork in Thailand. The authors also examined the association of AMR patterns with genotypic profiles. 

Overall, this is a well-written manuscript with a clear objective. I have a few comments

Few comments:

  1. In Table S2, The authors should highlight the AMR pattern [R, I, S] with different colours.
  2. Table 2 and Figure 2 should be supplementary. 

Author Response

Thank you for your valuable comments. Please find responses below.

Comment 1: In Table S2, The authors should highlight the AMR pattern [R, I, S] with different colours.

Response 1: We revised the table, accordingly. In addition, we put colors to Table 2.

Comment 2: Table 2 and Figure 2 should be supplementary. 

Response 2: We feel these are essential in main text. Sampling location is very important information for this manuscript. In addition. Basic data for SCCmec type and ST type are in supplementary tables and Table 2 includes just essential information for readers.

Reviewer 2 Report

In the present paper, the Authors investigate the epidemiology and molecular characteristics of MRSA in individual slaughter pigs and pork in Thailand.

The manuscript is clearly presented.

The objectives of the study are of interest and are perfectly in line with the scope of the journal. The introduction has enough information about the subject of the study. The methodology is not reported. The results reported are pertinent, and discussion sufficiently discusses the results.

However, in considerations of lacking of the description of material and methods, major revisions are necessary. Please describe this part.

As is, the manuscript is not acceptable for publication in Antibiotics.

Author Response

Thank you for your important comments. Please find responses below. We hope our manuscript has been improved much by your guidance.

Comment: In considerations of lacking of the description of material and methods, major revisions are necessary. Please describe this part.

Response: According to your comment, we added descriptions to material and methods. Please find them at Lines 289-299 and 302-307.

In addition, grammatical errors and typos were removed as much as possible to improve the manuscript.

Reviewer 3 Report

Very good analysis, just a few corrections in terms of English leanguage and phrasal editing such as:

Line 46: resistance instead of resistant

Line 49-50. The statement needs reference

Line 58: the phrase needs editing

Author Response

Thank you for your thoughtful advices. We revised accordingly. We hope our responses meet your points well.

Comment 1: Line 46: resistance instead of resistant

Response 1: We revised accordingly. Please find it with green highlight at Line 46.

Comment 2: Line 49-50. The statement needs reference

Response 2: We added a reference, accordingly. Please find it at Line 51 with green highlight.

Comment 3: Line 58: the phrase needs editing

Response 3: We revised accordingly. Please find it with green highlight at Line 58-59.

In addition, grammatical errors and typos were removed as much as possible to improve the manuscript.

Round 2

Reviewer 2 Report

Dear Authors,

the manuscript has been considerably improved. You have to insert the section of materials and methods after the discussion. The sections should be arranged as follows: 1. Introduction; 2. Results, 3. Discussion; 4. Materials and methods; 5. Conclusions.

The manuscript could be accepted for publication in Antibiotics after minor revision.